# Twelve Years into Genomic Selection in Forest Trees: Climbing the Slope of Enlightenment of Marker Assisted Tree Breeding

**Dario Grattapaglia** 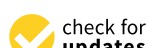

Plant Genetics Laboratory, EMBRAPA Genetic Resources and Biotechnology, Brasilia CEP 70770-970, DF, Brazil; dario.grattapaglia@embrapa.br

**Abstract:** Twelve years have passed since the early outlooks of applying genomic selection (GS) to forest tree breeding, initially based on deterministic simulations, soon followed by empirical reports. Given its solid projections for causing a paradigm shift in tree breeding practice in the years to come, GS went from a hot, somewhat hyped, topic to a fast-moving area of applied research and operational implementation worldwide. The hype cycle curve of emerging technologies introduced by Gartner Inc. in 1995, models the path a technology takes in terms of expectations of its value through time. Starting with a sudden and excessively positive "peak of inflated expectations" at its introduction, a technology that survives the "valley of disappointment" moves into maturity to climb the "slope of enlightenment", to eventually reach the "plateau of productivity". Following the pioneering steps of GS in animal breeding, we have surpassed the initial phases of the Gartner hype cycle and we are now climbing the slope of enlightenment towards a wide application of GS in forest tree breeding. By merging modern high-throughput DNA typing, time-proven quantitative genetics and mixed-model analysis, GS moved the focus away from the questionable concept of dissecting a complex, polygenic trait in its individual components for breeding advancement. Instead of trying to find the needle in a haystack, i.e., the "magic" gene in the complex and fluid genome, GS more efficiently and humbly "buys the whole haystack" of genomic effects to predict complex phenotypes, similarly to an exchange-traded fund that more efficiently "buys the whole market". Tens of studies have now been published in forest trees showing that GS matches or surpasses the performance of phenotypic selection for growth and wood properties traits, enhancing the rate of genetic gain per unit time by increasing selection intensity, radically reducing generation interval and improving the accuracy of breeding values. Breeder-friendly and cost-effective SNP (single nucleotide polymorphism) genotyping platforms are now available for all mainstream plantation forest trees, but methods based on low-pass whole genome sequencing with imputation might further reduce genotyping costs. In this perspective, I provide answers to why GS will soon become the most efficient and effective way to carry out advanced tree breeding, and outline a simple pilot demonstration project that tree breeders can propose in their organization. While the fundamental properties of GS in tree breeding are now solidly established, strategic, logistics and financial aspects for the optimized adoption of GS are now the focus of attentions towards the plateau of productivity in the cycle, when this new breeding method will become fully established into routine tree improvement.

**Keywords:** genomic selection; forest tree breeding; SNPs; MAS

## 1. Introduction

Almost 12 years have passed since the early prospects of applying genomic selection (GS) to forest tree breeding, starting with deterministic simulations [1,2], and quickly followed by empirical reports in *Eucalyptus* [3], *Pinus taeda* [4,5] and *Picea glauca* [6]. A quick perusal of Web of Science on July 20, 2022 using the phrase "genomic selection" and the individual keywords "forest", "tree" and "breeding" returns 81 papers when searching in the field tag "Topic". Given the predictions for causing a paradigm shift in the way that tree breeding will be done in the years to come, genomic selection has definitely become a

hot topic in forest tree breeding and a fast-moving area of applied research and operational implementation in several organizations worldwide, both public and private.

At least two extensive books have been published with an in-depth treatment of genomic selection applied to plant breeding [7,8]. A number of reviews have also been published, describing the fundamental aspects of genomic selection specifically applied to tree breeding, following the incremental advances made in the area [9–15]. Given the abundance of recent and detailed reviews, it would be pointless to reiterate once again such review style presentation. Moreover, in the interest of a smoother reading, I largely refer the reader to those reviews to support the technical aspects of my discussion, although some original papers are cited whenever necessary.

In my encounters with tree breeders and forest production practitioners, several questions about GS are posed, but in general their true objective is to ask: is GS a real thing or just one more hype, one more biotech bandwagon [16] passing by? Should I hop on or just let pass and move on with my current methods? How should I start a GS program? In this piece, I attempt to provide my perspectives on why GS will soon become an efficient and effective way to increase genetic gain per unit time in tree breeding programs. This is done by providing answers to a few frequently asked questions, in which I attempt to cover some aspects that deserve additional reflections when contemplating the adoption of GS in a tree breeding program. This is followed by outlining the steps of a simple, accessible, pilot demonstration GS project that should be adaptable to the biology of different tree species. Evidently, however, the final consideration whether to embark in this alternative breeding approach will depend on a number of additional variables that space will not permit dealing with here but that can be found elsewhere [10,12]. It is important to point out from the start that I will address some of these most frequently asked questions from my personal perspective and experience, which is unavoidably patchy in coverage and references, and reflects my interests, biases, knowledge and gaps.

## 2. Climbing the Slope of Enlightenment of Marker Assisted Forest Tree Breeding

The hype cycle curve, introduced by Gartner Inc. in 1995 (accessed on 22 July 2022 https://www.gartner.com/en/documents/3887767), models a largely applicable path a technology takes in terms of expectations or prominence of the value of the technology (y-axis) through time (x-axis). It is formed by merging two distinct curves that are able to explain the hype curve shape of new technologies. The first curve corresponds to a sudden overly positive and sometimes irrational reaction on the introduction of a new technology that involves attraction to novelty and heuristic attitude in decision making. This has happened with some of the ever-growing bandwagon biotechnologies that end up falling into a valley of disappointment [16]. For those technologies that survive, this initial phase is then followed by a second s-shaped curve that corresponds to technology maturity based on the notion that the performance of the technology develops only slowly in the beginning. Its fundamentals are poorly understood and investments into pilots and early adoptions may result only into little performance gains. Depending on the technology, at some turning point the technology performance is supposed to take off until a plateau, defined by the technology's specific limits, is reached. The timeline of marker assisted breeding in forest trees closely tracked the Gartner hype cycle through its different phases along the last 30 years (Figure 1). In trying to accommodate the timeline of marker assisted tree breeding with the different phases of the hype cycle, some leeway on the reader's part is necessary. Some events are placed along positions of the cycle that do not correspond to the exact timeline or, worse, to the specific cycle phase. For example, the advent of accessible high-throughput sequencing and genotyping technologies definitely does not belong to the disillusionment phase!

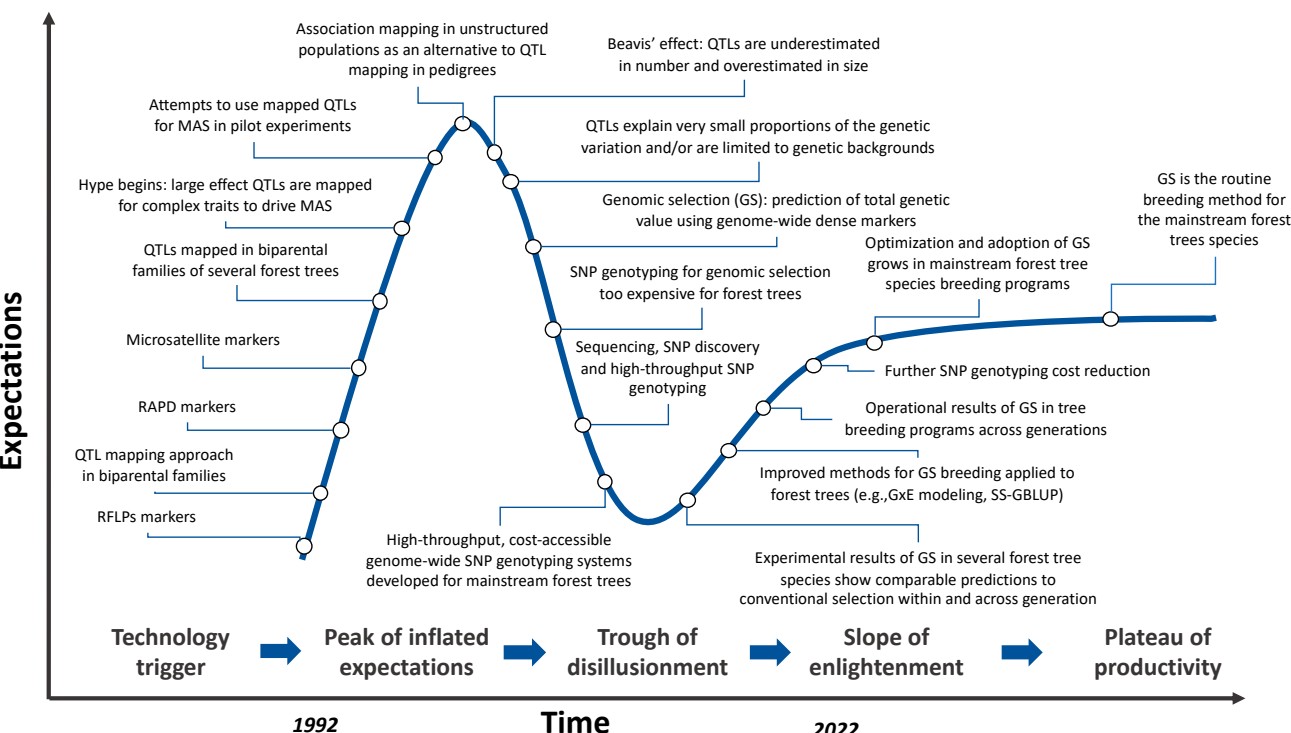

**Figure 1.** The timeline of marker assisted breeding in forest trees closely tracked the Gartner hype cycle through its different phases along the last 30 years. To accommodate the timeline with the hype cycle, the sequence of events not always reflects their exact relative chronology along the curves or fit the specific phase of the cycle.

Following trigger technologies, including methods to detect polymorphisms in DNA, and QTL (quantitative trait locus) mapping in biparental crosses, MAS (marker assisted selection) was proposed as a way to dissect complex trait, carry out early selection and thus accelerate tree breeding [17,18]. A number of mapping studies soon followed, and seemingly major effect QTLs were mapped in several forest tree species [19–21], results that further hyped the curve, leading to a peak of inflated expectations. Well-grounded doubts had been casted, however, on the prospects of MAS in outbred trees [22], and results from simulation studies in crops indicated that QTLs (quantitative trait loci) were likely vastly overestimated in effect size and underestimated in numbers [23]. Based on improved sequencing and genotyping methods, association mapping came into the scene as the solution to MAS in forest trees [24], further feeding the expectations of dissecting complex traits in individual components. However, pilot experiments to use mapped QTLs for MAS, largely unpublished, failed, as the QTL effect sizes were irrelevant or unreliable across populations, driving the field into a trough of disillusionment. GS had already been proposed in 2001 [25], at the same time as all these movements were happening, but only animal [26] and crop [27] breeders had figured that it was the effective solution to marker assisted breeding. We introduced GS for forest trees in a review paper as the likely answer to the promise of using DNA marker data to advance breeding, in contrast to the trait dissection approach [28]. With the initial simulations and a number of empirical reports, the field of GS in tree breeding took off [9], and has now quickly evolved [15]. This evolution, together with the solid basic principles underlying GS, make me confident to argue that we are now climbing the slope of enlightenment of the now 30-year-old Gartner hype cycle of marker assisted forest tree breeding.

## 3. Why Is GS Any Different from MAS?

Many professionals in forestry might not have had a chance yet to fully capture the main aspects underlying the basic idea of genomic selection [25] compared to what was originally proposed and still practiced today as marker assisted selection (MAS) [29]. The purpose of

MAS and GS is common: using DNA markers as indirect selection criteria for qualitative or quantitative trait loci controlling agriculturally important traits for improved breeding efficiency, including reduction of breeding cycle length, phenotyping cost reduction and higher selection precision and intensity. Although the ultimate goals of MAS and GS are the same, and sometimes GS is classified as a variant method of MAS, they have important differences that need to be emphasized for a better appreciation of GS. I start by addressing this point, comparing the fundamental premises and underlying concepts that differ between these two approaches (Table 1). Furthermore, GS is sometimes inaccurately classified as a biotechnology or a genomic technology sensu stricto [30]. In reality, GS is a breeding method, one more tool in the toolbox of the breeder, like the several tools successfully used to date, such as the theory and practice of experimental designs, quantitative genetics, mixed model analysis, BLUP (best linear unbiased prediction), hybrid breeding, propagation and flower induction methods, to mention a few in the case of plant breeding. This is why, in my opinion, GS should always stay in the responsibility of the breeding team and not as a separate area and much less in the "biotechnology" team of an organization.

**Table 1.** Comparative summary of the underlying concepts, philosophy, premises and practice of Marker Assisted Selection versus Genomic Selection.

| | Marker Assisted Selection | Genomic Selection |
| --- | --- | --- |
| Common purpose | Using DNA markers as indirect selection predictors of quantitative trait loci controlling agriculturally important traits in unphenotyed individuals, for improved breeding efficiency, including reduction of breeding cycle length, phenotyping cost reduction and higher selection precision and intensity. | |
| Underlying historical concepts | Mendelian, discrete units of inheritance, known numbers and effects | Fisherian, continuous variables, infinitesimal model, unknown numbers and effects |
| Underlying philosophy | Largely reductionist: complex traits are amenable to dissection in its individual component loci, and experiments should be sufficiently statistically powered to detect and discretely identify them | Holistic: traits are complex, multifactorial, i.e., controlled by a large unknown number of loci and power is never sufficient to discretely identify the effects of all of them |
| "Needle and haystack tactic" | Based on the concept of the existence, finding and use of major effect "needles" (QTLs) in the "haystack" (genome) | Traits are polygenic and rarely, if ever, there are major effect "needles"; it is safer, humbler and cheaper to "buy the whole haystack", i.e., capture the whole genome effect |
| Statistical approach | Statistical inference: given a set of co-segregating marker and phenotype data, aims at understanding the relationship between the predictors (markers) and the output variable (phenotype) | Statistical prediction: given a set of marker and phenotype data, models are trained that more reliably predict an outcome (phenotype) that is yet to happen, based on predictors (markers) |
| Hypothesis testing | Based on (stringent) hypothesis testing, interested in establishing the presence, location and magnitude of an effect (QTL) in a marker interval in the genome | Does not use hypothesis testing; not interested in determining the presence, location or magnitude of an effect |
| QTL detection | Exploits only those few QTLs that reached the significance threshold in the prior QTL mapping experiment; other QTLs remain undetected | Does not count on any prior QTL mapping detection |
| QTL effect | QTLs used are frequently overestimated due to the Beavis effect (i.e., QTL sampling from a truncated distribution in a limited sample of individuals) | Generally, not interested in determining the magnitude of individual QTL effects (although feature selection, i.e., selection of a subset of predictor markers sometimes used) |
| QTL number | QTLs used are frequently underestimated in number due to the Beavis effect; small effect QTLs go undetected due to limited power of prior QTL mapping | Not interested in knowing how many QTLs are controlling the trait; the theoretical infinitesimal model works well to capture most large and small effects on the target trait |
| Sequence of events in breeding practice | Establish significant marker-QTLs associations, validate them (sometimes), and then select individuals based on the presence of these discrete "a priori" known QTLs | Precludes the prior search for QTLs and focuses on prediction of breeding or genotypic values of individuals based on past data from a training population |
| Use of phenotypic information from relatives | Information comes exclusively from prior QTL mapping experiments whether involving individuals related or unrelated to the current selection candidates; no explicit use of genomic relationships | Uses all genetic and phenotypic information from relatives in prior and parallel generations, through genomic relationships |
| Use of external information | Sometimes uses prior information from molecular biology studies to search for QTLs or seek external QTL validation | Agnostic to any prior (molecular biology) mechanistic information although it can be incorporated into the prediction model, if desired |
| Selection system | Targeted, optimized, based on discrete markers with previously estimated and expected effects; higher risk of failure due to system overoptimization with low generalizability | Markers distributed across the genome, with marker redundancy and uncertainty built in the reliability of the prediction model; low risk of system overoptimization |
| When it works | Typically works for traits displaying major effect, high penetrance loci/variants in homogeneous (low diversity) genetic backgrounds (e.g., crops) shared by the QTL mapping population and the MAS candidates | Works for any trait architecture in any genetic population structure, as long as model training population is genetically related to the selection candidates and the effective population size stays within $N_e \approx 100$ |

### 3.1. GS Is Based on Fisher's and Wright's Methods with DNA Markers

In a nutshell, GS is a biometrical method to predict the genetic merit of a genotyped but unphenotyped individual tree of family, typically called selection candidate, based on prediction equations built from a large ancestral population genetically related to the selection candidates, called training or calibration population, for which both phenotypes and genotypes are available (see [15] for illustrative charts). The main difference from conventional pedigree-based methods is the use of genome-wide DNA marker data, usually SNPs (single nucleotide polymorphisms), which improve the estimate of relationship among individual trees and, in doing that, also improve the prediction of breeding or genotypic values. As Ignacio Misztal [31] straightforwardly put it, "*think of genomic selection as the animal model with more accurate relationships*". Reminding that the term "animal model", a particular case of a mixed model, comes from its extensive use in domestic animal breeding.

So the first key aspect of GS is that, in essence, it is based on the same principles and methods invented by Fisher and Wright in the beginning of the last century involving the infinitesimal model and the resemblance between relatives [32,33], and further developed by Cockerham [34] and Kempthorne [35]. The difference is that DNA marker data captures more granular genetic relationships between individual trees, when compared to the recorded pedigrees, allowing more accurate predictions of breeding values for the unphenotyped selection candidates. It then follows that one is able to rank trees on their predicted performance based exclusively on their DNA marker information. Being able to use DNA data to accurately predict the future performance of a tree, *per se* as a clone or as a breeding parent, when it is still a young seedling with no growth, form and wood phenotypes, is the core attractiveness of genomic selection in forest trees and long-lived perennial species in general.

### 3.2. From Inference to Prediction

The second key aspect of GS, is that while MAS is based on statistical inference, GS is based on statistical prediction. Both procedures learn from the data to find a model that describes the relationship between the independent variables (the markers) and the outcome (the phenotype). They however diverge when it comes to the objective use of the resulting model. In MAS, on the basis of the fitted model, one wishes to interpret the role of the marker-associated QTL on the measured phenotype. In GS one fits several models using all markers and phenotypes on the training data and selects the model that better predicts the future phenotypes following cross-validation. MAS is based on the concept of dissecting a quantitative trait in its individual components and using them in breeding. GS on the other hand, is based on capturing the whole genome effects of the large numbers of variants of Fisher's infinitesimal model, a model that has provided a very good approximation to reality. In the era of high throughput and inexpensive genome sequencing, this same reductionist approach of MAS has been extended to the proposal of discovering individual genes, regulatory variants, or the actual QTN (quantitative traits nucleotide), and decipher gene networks for complex traits. Despite some occasional advances, mostly in crops, the results of such efforts remain in the realm of promises in what concerns applied tree breeding. The risk of such a challenging approach is that the final number of elements and their interactions in such networks is so large that from the practical standpoint, it will closely match the infinitesimal model as a proxy, anyway.

The underpinning of the marker assisted selection approach is the "major effect" QTLs, which were proven to be largely overestimated in effect size, underestimated in number [36] and rarely, if ever, independently validated [37,38]. Applying the "major effect QTL" paradigm from inbred crops to breeding and "gene discovery" in forest trees unfortunately has been deceptive, wasting time, money and effort. The very recently domesticated forest tree populations still closely behave as wild populations where the fundamental evolutionary forces operate on the micro-mutational architecture of quantitative traits [39]. This is quite different from annual crops that have been the subject of directional selection

for centuries or even millennia. While the macromutational model of major effect variants at intermediate frequencies behaving as Mendelian genes has delivered some QTL mapping and GWAS success stories in annual inbred crops, it has found no room yet in forest trees. Such large effect variants are usually caught by simple observation in annual plants, and later subject to genetic mapping and molecular dissection. This same practice is difficult if not impossible for the relevant phenotypes in large long lived perennial forest trees.

The futility of explaining genetic variation for quantitative traits in terms of the underlying individual components through linkage mapping and GWAS [40] for what regards breeding advancement, and hence the inherent limitations of MAS, has been recognized in domestic animals [41], crops [42] and forest trees [28] for some time now. Except for those rare cases of obvious major effect QTLs, the "philosophy" of trait dissection in individual components is therefore rapidly becoming obsolete. Given that genome-wide marker technologies are readily accessible, the MAS approach as outlined in Table 1 will be rapidly absorbed by the whole genome approach of GS, simply because GS is more efficient as it takes into account the effect of the entire genome. Additionally, whenever major effect QTLs with strong validated genetic signal are known, they can be integrated into GS models, potentially improving their prediction ability as shown in crops [43].

I therefore contend that, after passing the MAS hype, we have now reached a consensus in the forest tree genetics and breeding community that the majority of, if not all, traits relevant to forest production fit Fisher's infinitesimal model involving a large number of mutations of very small effect, however unrealistic this model may be. *"Remember that all models are wrong; the practical question is how wrong do they have to be to not be useful"* [44]. It follows that in the same manner as quantitative genetics, in light of the assumed complexity of the gene action underlying continuous traits, GS also qualifies blatantly as a 'black box' method and, by the way, there is nothing wrong with that.

## 4. What Are the Main Advantages of GS as a Breeding Method in Forest Trees When Compared to Conventional Breeding?

Discussing the advantages and expected impact of GS in tree breeding is simplified by using Falconer's breeder's equation as the conceptual framework [45]. Although the genotyping cost does not directly impact the response to selection, it is frequently seen as a cost limiting factor for adopting GS in a tree breeding program. I therefore included it as an additional denominator term that needs to be considered. The breeder´s equation, now a genomic breeder´s equation, then becomes $\Delta G = ir\sigma_A/Lg$ where $i$ is the selection intensity (the proportion of trees that are selected to be clonally propagated or used as parents of the next generation); $r$ is the accuracy of selection, i.e., the correlation between the estimated breeding value (EBV) and the true breeding value; $\sigma_A$ is the additive-genetic standard deviation of the trait of interest; $L$ is the breeding cycle length; and $g$ is the genotyping cost per sample. For a fixed amount of starting genetic variation available in the breeding population ($\sigma_A$) and a fixed (and as low as possible, given the necessary marker density and quality) genotyping cost per sample ($g$), GS will increase the rate of genetic gain per unit time by decreasing the denominator and increase the numerator terms that remain. The addition of the genotyping costs of GS in the breeder's equation should however be seen just as an unpretentious practical proposal. It does not represent an exact comparison to conventional breeding. Conventional phenotype-based selection will have costs that might be precluded with GS, namely, (1) the phenotyping costs incurred in each generation that are avoided with GS and (2) the opportunity costs derived from the time saved in accelerating breeding cycles. So, for a better representation of the ($g$) term, the two breeding strategies should to be balanced against each other by their ratio. Because these costs vary a lot across species, traits and breeding programs this exercise should be done on a case-by-case basis.

*4.1. Shorten Breeding Cycles and Increase Selection Intensity*

The first, most obvious and likely the foremost advantage of GS, when compared to conventional breeding in forest trees, is the ability to reduce (*L*) by carrying out very early selection of unphenotyped seedlings in the nursery, while simultaneously increasing selection intensity (*i*) by selecting among many more seedlings. For example, in *Eucalyptus*, a typical progeny trial involves several tens of half or full-sib families with tens of individual trees per family for a total of a few thousand individual trees tested. This requires large experimental areas and at least 4–6 years-time to adequately measure growth and especially late expressing wood traits. GS allows saving time by precluding the traditional field progeny trial by very early testing of many more families and trees per family, better exploiting the between and especially the larger within-family genetic variation. Genomically selected seedlings, once subject to early flower induction by top grafting and chemical treatments [46], could be recombined to create the next generation, therefore shortening the breeding cycle and increasing genetic gain per unit time [15].

GS is most frequently seen as a way to find the "winner" trees. However, culling the losers might be equally valuable from the operational standpoint, reducing testing costs and optimizing resources in the program, by advancing a smaller pre-selected number of trees to final field performance testing. This has been very valuable, for example, in dairy cattle GS programs by culling inferior females at young age [47]. For programs that employ family forestry or clonal forestry, the genomically pre-selected trees in the nursery would be clonally propagated and multi-site field tested as clones. Using multi-trait genomic prediction models, early selection would be carried out for multiple traits simultaneously in large numbers of individuals, an almost impossible task in conventional tree breeding that largely adopts tandem selection. In conifers, where clonal propagation by somatic embryogenesis is an option, genomically selected seeds could be immediately placed in tissue culture, accelerating the production of embryogenic lines and reducing the costs and uncertainties of embryo cryopreservation and rescue [4].

*4.2. Increase Accuracy of Predictions and Genetic Parameters*

The second expected advantage of the incorporation of genomic data in the prediction of genetic merit is to increase the selection accuracy/reliability (*r*) of the estimated breeding values and genetic parameters when compared to conventional selection using exclusively pedigrees [48–50]. Unlike the average numerical relationship matrix based on the expected pedigree, the realized genomic relationship matrix built from SNP data finely reflects the actual fraction of the genome that is identical by descent or by state between individuals within-family, resulting from the random Mendelian segregation Additionally, SNP data detects the obscure relationships among individuals across families, and corrects unknown pedigree inconsistencies in the trial [51,52]. By increasing the accuracy in estimating the additive variance ($\sigma_A$), heritability and breeding values, GS ultimately provides a more precise and realistic estimate of genetic gain. Still, in relation to the ability of genomic data to reconstruct pedigrees, this could be particularly useful when controlled crosses are difficult or expensive to make and breeding is carried out exclusively by an open pollinated or polymix strategy with exclusive maternal control. The SNP data in effect could convert the program to a fully pedigreed one, albeit likely unbalanced.

The realized genomic relationships can also be used in combination with the pedigree-based average relationships of a much larger set of phenotyped but ungenotyped trees in the trial. This approach called single-step GBLUP (genomic best linear unbiased prediction), SS-GBLUP also called HBLUP, allows using phenotypic information from the entire trial, increasing the precision of the genetic parameters estimated from conventional pedigrees, with the benefit of limiting overall genotyping costs. This approach has been particularly useful in forest trees where budgets for genotyping costs are usually limited, and trials may involve a large number of individuals [53,54].

## 5. What Factors Impact the Success of GS?

Several are the factors that determine the success of a genomic selection program. As I outlined earlier [10], these can be divided into two groups. The first group involves the four key factors from the theory of population and quantitative genetics that affect the accuracy of genomic prediction. The second group involves the practical aspects of cost benefit analysis and resource allocation in a tree breeding program. We have now reached a point where some general conclusions can be drawn and recommendations made as to what exactly are the most relevant issues defining the success of GS.

### 5.1. Key Factors from the Theory of Population and Quantitative Genetics

This group involves: (1) the effective size ($N_e$) of the reference population and genotyping density that jointly govern the number of independently segregating chromosome segments, which in turn determine the extent of markers-QTLs linkage disequilibrium and family relationships; (2) the size and composition of the training population, i.e. the number of individuals with phenotypes and genotypes used to develop the predictive model; (3) the heritability of the trait in question; and (4) the genetic architecture of the target trait, i.e., the distribution of QTL effects (number of loci and size effects) [55]. Deterministic simulations were used to assess the influence of each one of these factors individually in the context of tree breeding, providing initial guidelines for GS regardless of species, recombinant genome size and breeding cycle length [1]. These factors were later examined experimentally, showing very good agreement with the simulations, as recently reviewed [10,12,13]. The first three factors are known and can be controlled experimentally to an extent, while the fourth is largely unknown but the infinitesimal model solves the issue well. In summary, given the typical sizes of reference populations in tree breeding ($N_e < 100$), GS training populations of several hundred up to 2000 individuals, and moderate marker densities (>5000 to 15,000 SNPs), experimental results in several forest trees species have unambiguously shown that the level of relatedness between the training population and the selection candidates is the main factor to impact the success of GS. The closer the relationships the higher the predictive ability.

This general conclusion was recently assessed statistically based on a compilation of 26 studies in forest trees by F. Isik [12]. He showed that the effect of relatedness (full-sib versus half-sib family structure) was highly significant on prediction accuracy for tree height while training population size and marker density were not significant. Increased relatedness reduces the number of independently segregating chromosome segments, therefore increasing the probability that chromosome segments identical by descent sampled in the training population are also found in the selection candidates. Furthermore, it has been shown that accumulated length of shared haplotypes across a reference individual and a selection candidate are more important in determining the reliability of genomic prediction than individual length of shared haplotypes [56]. Given the very recent history of forest tree domestication (now barely in the third to fourth generation of breeding), and the very large and diverse genetic base of the worldwide germplasm for any single species, breeding populations differ widely in allele frequencies, marker-QTL linkage disequilibrium (LD) phase, haplotypes and population-specific allele effects. Predictions across breeding populations or in the absence of family relationships, have been shown to be very low or nil [3,57]. Family relationships within any single population, provided that the effective population size stays within $N_e \approx 100$, will thus represent the key driver of GS prediction accuracy in forest trees. For the successful implementation of GS, it is therefore absolutely crucial that the foundational population used to train the genomic prediction model must represent the genetic diversity and be genetically related the selection candidates as best as possible. This conclusion is currently seen as a consensus in the field, which in turn determines that GS will be breeding-population specific, requiring that each organization develops a predictive model for its reference genetic base. This becomes even more true when the genotype by environment factor is taken into account. The idea of a universal GS model applicable across populations seems, therefore, as unrealistic as the similarly

utopian idea of discovering individual genes, genomic elements or QTLs for polygenic traits that will work for directional selection across populations.

### 5.2. Practical Factors of Resource Allocation in a Tree Breeding Program

The second group involves the several aspects that define costs and resource allocation in the breeding program: (1) the phenotyping effort devoted to the training population (cloning individuals, how many traits to measure and model, cost of measurements); (2) genotype by environment (GxE) and age interaction, multi-trait selection (single or multiple environments and trait GS model(s)) (3) SNP genotyping system (data quality, cost, turn-around time and breeder's friendliness); and (4) expected long term performance of the GS model (effort devoted to model retraining, control of inbreeding). These factors involve several organization-dependent aspects and a thorough discussion would be beyond the scope of this article but more detail can be found [10,12]. Some of the main aspects are summarized below.

### 5.2.1. Phenotyping of the Training Population

In principle, as much effort as possible should be devoted to generating the best phenotypic measurements for all relevant traits on the individuals that make up the training population, because these measurements will directly impact the quality of the predicted trait values of the unphenotyped selection candidates. Moreover, several experimental evidences from crop plants [58] and forest trees [59,60] have shown that multi-trait models improve prediction accuracies of breeding values. When the information of trait correlation is incorporated in the analyses, low heritability traits may benefit when analyzed in conjunction with traits showing higher heritabilities. These results further support the importance of dedicating considerable efforts to phenotyping the training population.

Typically, a progeny trial and its parents have been used as the training population. Whenever biologically feasible, the individuals of the training population should be clonally propagated. Cloning provides important additional accuracy in individual BLUP evaluations of the training individuals especially for lower heritability traits [61]. In other words, efforts should be dedicated to increasing heritability and using the most reliable phenotypic values possible for the individuals used to train the genomic prediction model. Additionally, in the case of vegetatively propagated species, where the target of GS not only is breeding value, but also genotypic value of the tree to be deployed as a clone, training models on clonal means is highly advantageous and should be pursued, especially for traits with low heritability and/or that would benefit from destructive sampling. Moreover, when the same reference breeding population can be used to breed trees for different environments, cloning will allow using the same genotype data on phenotypes collected on the clonally replicated individuals collected in the different environments and different environment-specific genomic models built for optimized predictive ability.

Finally, while genotyping has become increasingly cheaper, phenotypic evaluation costs have frequently increased with raising labor costs. Different strategies for the optimization of the training population have been proposed as well as different algorithms and software tools [62]. Training set optimization consists of selecting from a reference population, typically a large progeny trial, a set of individuals to be phenotyped that are expected to provide better genetic representation and relatedness to predict future unphenotyped candidates. Selective phenotyping strategies can be used in which phenotyping is devoted preferentially to individuals to maximize genetic dissimilarity. A random sampling of individuals to build the training population might be risky because it could lead to low-quality coverage of the total genetic space, especially when the reference population contains population structure [63]. Optimized allocation of resources ultimately aims at reducing costs of phenotyping and increasing the quality of the phenotypic data by focusing on traits that are more expensive or difficult to measure, or increasing complementary measurements of the same traits.

### 5.2.2. SNP Genotyping Platform Data Quality and Cost

All published studies in forest trees have shown suitable predictive abilities using moderate genotyping densities with little or no gain above 10,000–15,000 datapoints, likely due to the prominent role of relatedness as the driver of predictions. Such polymorphic marker numbers are now comfortably provided by the several medium-density, mostly public, SNP arrays with 30,000 to 70,000 SNPs for all the mainstream plantation forest trees species. These platforms were initially developed through individual project for *Populus* [64] and *Picea* [65] followed by community-style organized efforts pioneered for *Eucalyptus* with the multi-species EuCHIP60K [66], upgraded to the Eucalyptus72K array in 2018 (accessed on 30 July 2022 https://www.thermofisher.com/order/catalog/product/55 1134), a model later reproduced for several conifers [67–72]. These SNP arrays are currently the best alternative as far as data quality, breeder's friendliness and genotyping cost, usually in the range of USD 22 to 30/sample. The assembly of large numbers of samples through community groups has been the key factor to drive genotyping cost reduction. Moreover, the easy access to shared commercial SNP arrays has been a major strength in advancing GS into operational use in forest trees, following the examples in domestic animals.

Because no gain in predictive ability has been generally reported when using more than 10,000–15,000 SNPs in predictive modes, one may ask, why not using lower density SNP arrays? The cost of manufacturing a SNP array largely depends on the prospective number of samples to be genotyped, rather than on the number of SNPs genotyped in the array. In other words, currently the sample genotyping cost reduction from a 60K SNP chip to a 10K SNP chip is minimal or likely none. Furthermore, minimum allele frequencies for the same SNPs frequently vary across provenances and species within the genus, so that SNPs will be more or less informative in different breeding populations. Moreover, simulations have shown that higher marker densities can keep GS effective for more generations and yield larger genetic gains when compared to using lower marker density [73,74]. Additionally, even in the absence of selection, recombination under low marker density more rapidly dissipates descent relationships when compared to higher marker densities, negatively impacting predictions [73]. Therefore, it is valuable or even necessary to have an excess of informative SNPs on the chip in the best interest of ensuring some redundancy while avoiding the risk of system over optimization.

### 5.2.3. Genotype by Environment Interaction and Age-Age Correlation

Genotype by environment (GxE) interaction is a fact of life for most if not all tree breeding programs. GxE can have different impact on the program, depending on the species, environmental variability and extent of the intended forest plantation sites and type of planting material, whether families, less interactive, or clones, more interactive. Interactions can be subtle when the relative ranking of tested trees does not change across environments (scale effect), or more severe when rank changes are observed. In forest trees, multi-environmental GxE interaction studies are regularly used to assess the performance of the same clones or families across different environmental conditions, to study genotype stability and to predict the performance of untested genotypes [75]. The need to develop specific GS models for each environment or breeding zone will largely depend upon the type of interaction observed, whether scale effect or rank change.

A second aspect to be considered when developing a genomic prediction model is the impact of the age at which the training population is measured on the accuracy of future predictions. Typically, the breeder already has robust data on juvenile-mature correlations, especially for wood properties, as it is common practice to make selections at an earlier age in an attempt to accelerate a breeding program. Moreover, the availability of repeated measurements on the training population allows including age in the model, potentially improving its prediction ability. Studies to date they have shown that data from traditional GxE or age-age correlation studies adequately inform what to expect from genomic prediction across environments and ages [12,13]. As a general rule, predictive abilities are reduced when models trained in one environment were validated or used to

predict phenotypes in a different one, although the magnitude of such reduction varied across traits. To achieve the most accurate predictions, models should be trained on traits measured in the same or equivalent environment and age as those where predictions on selection candidates will be made. As age-age correlations between training and testing age improve, and the magnitude and trend of the GxE interaction becomes inconsequential between training and testing sites, predictions will tend to be satisfactory, provided that genetic relationships between training and selection candidates are kept in the population [10]. Below, when outlining some of the future perspectives in GS I will briefly mention a number of developments made in recent years on how to deal and exploit genotype by environment interaction to improve genomic predictions.

5.2.4. Long Term Performance and Sustainability of Genetic Gain with GS

We are only 12 years into genomic selection in forest trees. For most, if not all forest trees species, a breeding cycle takes longer than that. Therefore, there are no experimental data yet on this topic. Experimental data assessing the performance of GS across multiple generations of tree breeding will take a few decades to be published, unless the breeding program still has access to a sufficient number of phenotyped trees from several ancestral breeding generations such that DNA can be obtained and training models backtesting be carried out. However, the basic concepts from population genetics can be used to make some predictions. The attributes of long-term recurrent selection in forest trees, naturally result in obstacles to maintaining effective GS over generations. As breeding generations advance, genetic recombination, selection and drift will erode linkage disequilibrium and progressively dissolve the relatedness between the original training population and the selection candidates, causing a reduction of prediction ability. As discussed above, this will take place faster under low marker density. Additionally, directional genomic selection across generations will also alter the genetic architecture of the trait, via changes in allele frequencies, and patterns of LD, potentially unfavorably impacting prediction.

Simulation studies have been published on this topic. In the seminal study of genomic selection, the decline of accuracy over generations was estimated at 5% per generation, getting smaller in later generations [25]. Several other studies evaluating the effect of directional selection and the structure and depth of the training population (reviewed in [76]) arrived to a consensus recommendation today: marker effects have to be re-estimated frequently in order to maintain accuracy of genomic predictions over generations. In forest trees, two studies specifically evaluated this issue, one for *Eucalyptus* [77] and another for a 60-year conifer tree breeding program [2]. Both studies clearly showed that for a sustainable GS program, prediction models have to be updated frequently and better predictions are obtained as data from more breeding cycles are included in the training data set.

A recommended GS model updating strategy is to always try to include new phenotypic data in the updated training model that are closer to the current breeding generation [2]. In each GS cycle, an additional and possibly diverse set of genotyped selection candidates is deployed in progeny trial, preferentially cloned for increased phenotype quality. From that point on, every year the prediction model is updated with the inclusion of phenotypic data of the extra subset of trees from previous generations. This updating strategy, although it requires growing and measuring trees every generation, not only is expected to keep the prediction ability of the model, but also contributes to the continuous verification of the genetic progress of the GS program.

Maintenance of genetic diversity for long-term sustainable genetic gain with GS is another issue that has been questioned by tree breeders. Although this is a problem common to conventional breeding, the faster effect of drift and selection applied by GS could, on one hand, result in faster unintentional increase of deleterious allele frequencies causing inbreeding depression, while on the other hand, result in a loss of favorable alleles. This requires strategies that balance gain with diversity, endure improvement for traits under selection, and keep diversity for future breeding. Daetwyler et al. [78], however, have shown that by capturing the Mendelian sampling term, placing an increased emphasis on

the individual rather than family information, the genomic-level resolution of GS increases differentiation among sibs, when compared to conventional BLUP selection, allowing, in principle, a better control of coancestry and the rate of inbreeding. A faster progressive reduction of response to selection with the successive cycles of GS breeding might therefore be more of an issue. Measures to mitigate this effect include periodical model updating and verification of performance along the GS cycles of breeding [2], as well as strategies to avoid truncation selection, optimize mating designs or combine selection criteria and preservation of genetic variation [79]. In any case, there is always the possibility of introducing novel genetic variation in the breeding population if that really becomes a problem. It is important to remind however, that this should be accompanied by genomic model updating to include the new sources of genetic variation to keep relatedness at an adequate level.

## 6. How Has GS Performed When Benchmarked against Conventional Pedigree-Based Breeding?

First it is necessary to consider how should GS success be measured. Another important reminder is that the phenotype is also only a predictor of the true breeding value and has an error variance associated to it, just like a GEBV (genomic expected breeding value). Two metrics have been reported in experimental studies evaluating the performance of GS in forest tree breeding. The predictive ability (PA), which is the correlation between GEBV estimated from GBLUP, and the observed phenotype; and the predictive accuracy, the correlation between the GEBV and the true breeding value. The prediction accuracy is however a scaled predictive ability obtained by dividing the predictive ability by the square root of trait heritability [80]. Because the heritability of a trait is very much population and experiment dependent, it has caused some complications when trying to compare the performance of GS across studies for the same traits. For this reason, the straightforward PA has become the preferred metric to determine the success of GS, at least in forest trees.

GBLUP estimates of predictive ability in forest trees have varied across species, traits and studies, and generally have been as good as or slightly better than ABLUP (the standard method for predicting breeding values using the A matrix of expected relatedness among individuals based on pedigrees) with a mean around 0.45 across 26 studies, as recently reviewed [12]. Important to remind that such estimates have been produced by training and cross-validating GS models within the same generation, except for one study in *Pinus pinaster* that showed even higher PA of 0.70–0.85 for growth traits when models were trained with multiple generations [81]. In a recent, still unpublished, study in hybrid *Eucalyptus grandis* x *E. urophylla*, we have obtained high realized predictive abilities above 0.8 for growth traits using single-step GBLUP across generations in a reciprocal recurrent selection (RRS) program. The training population included phenotypes and genotypes of the prior generation hybrid progeny trial, as well as phenotypes of all trees in the progeny trials of the two pure species wherefrom the parents were selected to be intermated. For these latter trials genotypes of only a few hundred individuals were included. These results are very exciting as they corroborate excellent perspectives to significantly accelerate a RRS program by GS.

Proper benchmarking of GBLUP against ABLUP in terms of genetic gain per unit time will however entail more than a single generation comparison. Even if a GS model does not reach the same predictive ability obtained by conventional breeding in a two-generation experiment, the fact that the length of the breeding cycle can be greatly reduced, and/or the selection intensity increased, will result in a significant rise of the genetic gain per unit time. So, a more accurate comparison would actually require measuring the accrued genetic gain across a few generations of GS breeding versus pedigree breeding, which evidently it is not readily possible. Dairy cattle breeding with a longer history of GS is probably the best example where such a comparative assessment has been made, showing significant improvements in genetic gains per unit time in a number of production and fitness traits [47]. In forest trees, simulations based on deterministic models [1] and experimental data [82,83] have shown that GS would have a tremendous impact in terms

of improved efficiency in genetic gain per unit time when breeding cycle length is reduced and selection intensity increased.

Two simulation studies for a conifer breeding program have been published so far, evaluating the predicted economic impact of GS over conventional phenotypic selection considering not only genomic-based breeding but also alternative deployment scenarios. Overall, both studies reached the same conclusion: that GS would significantly increase genetic gain when compared to current breeding methods. Li and Dungey [84] simulated combinations of conventional or GS breeding with progeny or clonal deployment, the latter through cuttings of somatic embryos. Their results showed that larger training populations, higher heritability traits, clonal deployment of GS selected somatic embryos, and top-grafting of GS selected parents for faster breeding cycles, would result in higher genetic gains. Chamberland et al. [85] compared five breeding and deployment scenarios, also involving GS or conventional pedigree breeding and seedlings, cuttings or somatic embryo deployment. The setups that included genomic selection resulted in the highest land expectation value. It is important to note, however, that results of both studies are for very long conifer breeding cycles where gains were derived mainly from shortening such cycles. These studies did not specifically run a full economic analysis including, for example, genotyping costs, although these will likely dilute and become irrelevant in such long breeding cycles. Additional economic evaluation studies are needed, including all costs involved, and for faster rotations species.

## 7. How Can I Start, How Much Will It Cost and How Long Will It Take to Run a Pilot GS Project in My Program?

A tree breeding program is usually complex, involving multi-year pipelines that manage different populations and overlapping cohorts of germplasm, delivering genetic material in the form of clones or improved seeds. Additionally, a forest-based enterprise involves long-term planning of land use and wood supplies where risk has to be minimized. Changing breeding strategy often is done stepwise so as not to disrupt the product development process. For these reasons, the potential implementation of a genomic selection program, in my opinion, should start by establishing a pilot, simple, relatively small scale, genomic selection project that would run in parallel to the main pedigree-based breeding. This would allow the breeding team to climb the learning curve on the several issues of integrating genomic data into the selection process. The execution and results of a small-scale pilot project would: (1) keep costs under control; (2) provide opportunities to identify particular components of the breeding and deployment strategy that would most likely benefit from the use of genomic data; (3) progressively build the necessary additional infrastructure, skills and personnel training to be ready when time comes to adopt GS as the main strategy; and (4) demonstrate concrete results to the top management of the organization that would justify the necessary budget allocations for a full transition to GS.

At least five components are strictly necessary as a starting point to avoid embarking on a GS project with the risk of "jumping the gun" on the opportunity of evaluating the method due to incompetent execution. First, a structured definition of the breeding objectives and strategy adopted by the program. Frequently, the breeding strategy and objectives reside exclusively in the breeder's head. Second, a clear definition of the reference breeding population targeted by the conventional long-term breeding program. As relatedness is the main driver of GS, to generate relevant results for the organization, the pilot program should be based on the same reference population. Third, effective phenotype measurement protocols for all relevant traits so that the best prediction models possible can be built from the training population. Fourth, selection based on estimation of BLUP through mixed-model analysis that handles unbalanced data and incorporates information from relatives. Fifth, competent nursery and field operations procedures and thorough data management, for maximum precision in pedigree and phenotype records. Knowledge of the procedures and technologies involved in DNA genotyping is not strictly necessary as this component is usually outsourced.

In a previous review I outlined a nine-point tentative roadmap of the main components necessary to plan GS implementation in the organization [10]. The discussion here will be more practical, providing an illustrated plan of a simple, yet informative pilot GS demonstration project that could be adapted to any particular species (Figure 2). The first step involves defining the training population of trees that are genotyped and phenotyped to develop and cross validate predictive GS models for later use. A training population involves at least the elite breeding parents with an effective population size ($N_e$) preferably <100, and one or more high-quality progeny trials preferentially of fully pedigreed families, derived from inter-mating them. The training population will include at least 2000 individuals but more is better. Individuals should be sampled between and within families to represent the entire spectrum of genetic diversity, although very low performing families and individuals should be left out.

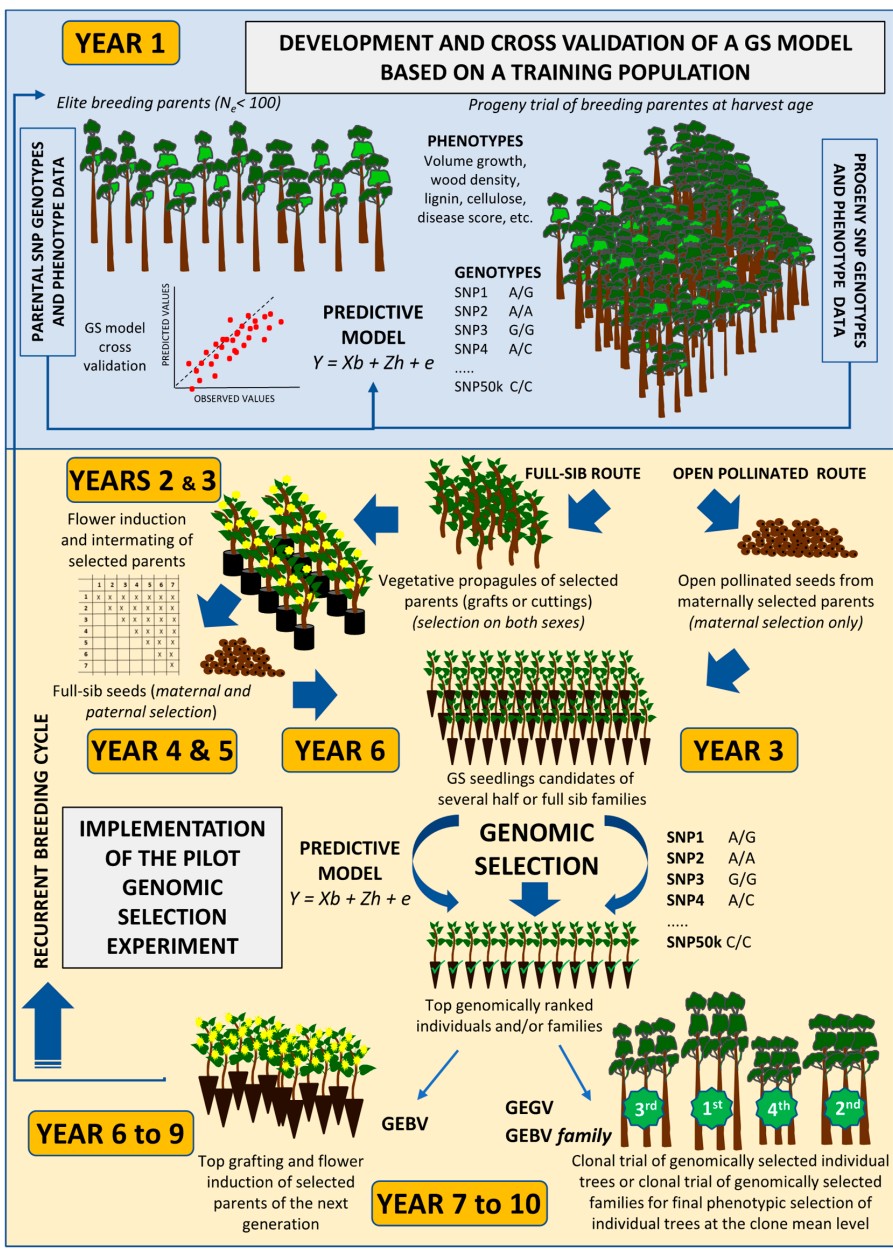

**Figure 2.** Step by step plan of a simple pilot project to evaluate and demonstrate the perspective of genomic selection in a recurrent selection breeding program. Timeline is suggested for a tropical *Eucalyptus* species, but should be adaptable to any other species (see text for details).

The training population will be genotyped with a SNP genotyping array, to generate in excess of 10,000 high-quality polymorphic SNP data points. If SNP arrays are not available for the species, a reasonably reliable SNP genotyping by sequencing can be obtained, for example, with DArTseq (Diversity Array Technology sequencing) [86]. The training population will be extensively phenotyped for all traits of interest. If possible, it is recommended cloning individuals of the training population so that models will be trained on the best phenotypic data possible and not individual tree data, although this will delay the actual implementation of the pilot GS project. Prediction models using genotype and phenotype data of the two generations (parents and progeny) will be built and cross-validated using a subsample of the training population. Considering the overall efficiency of GBLUP, a general recommendation has been made to use this method as a starting point.

Once a satisfactory prediction model has been developed, it will be used on the GS candidates for forward selection in this example. GS candidates will be an array of half or full-sib families derived from intercrossing selected individuals in the progeny trial that took part of the training population. For a faster advancement of the pilot project, forward selection can be carried out only on the female side by collecting open-pollinated seeds in the trial. For higher genetic gain (selection on both sexes) but slower project progress, vegetative propagules of selected trees in the trial are collected, propagated, either by grafting or cuttings, and flower induction carried out. These selected parents will be intercrossed to generate fully-pedigreed GS candidates.

At this point depending on the final deployment of improved genetic material, different routes can be followed. If final deployment is by seeds only, seedling candidates are genotyped and their genomic breeding values (GEBV) estimated using the predictive model developed earlier. Genomically selected seedlings are top grafted in the breeding orchard and inter-mated to produce improved seeds for planting and selection candidates for the next breeding generation. If final deployment of improved genetic material is by clonal propagation, the route will depend whether the track record of the breeding program has shown that the individual tree performance in a progeny trial is a good predictor of its performance as a clone. This has been shown not to be always the case in the few studies that examined this issue in tropical hybrid eucalypts, for example [87,88]. If the correlation of individual performance versus its clone is high, the genomic expected genotypic values (GEGV) of the candidates are estimated and selection practiced using a model that takes into account non-additive effects. To assess the realized predictive ability of the GS project, these selected individuals are clonally propagated and deployed in clonal trials.

However, when the individual tree performance in a progeny trial is not a good predictor of its performance as a clone, a two-stage selection is recommended. GS would be applied only at the family mean level in the first stage, where predictive ability is typically higher and more reliable [89], and conventional phenotypic selection applied in the second stage at the individual tree level while assessed as a clone. In this case a larger number of families should be considered for higher selection intensity in the first stage, as only the between-family additive variance is exploited by GS in this scenario. Only a sample of individuals per family or even only a few replicate pools of individuals per family [90] need to be genotyped in a family pooling approach, significantly saving on genotyping costs. GEBV is estimated at the family mean level with the appropriate prediction model. The top ranked families would have several individual seedlings cloned and deployed in a combined cloned progeny trial for higher selection intensity in the final phenotypic selection. This combined progeny-clonal trial strategy saves considerable time when compared to the conventional path that involves separate and sequential progeny and clonal trials.

Irrespective of the route taken, besides the selected individuals, a random subset of the already genotyped selection candidates should also be cloned and planted in the trial to serve as controls, and to provide additional data for GS model updating. I am not considering here the possibility of using muti-environment modeling to account for

genotype by environment interaction and neither multi-trait selection, although both factors could evidently be included if desired.

Regarding direct costs to implement such a pilot GS project I will assume that the training population is available and phenotyped. Costs of seed collection, seedlings production and data analysis can be easily encompassed in the routine activities of the breeding program, and will not be considered. Thus, the costs would involve DNA extraction and genotyping of the 2000 trees of the training/validation population and a sample of selection candidate seedlings (let's assume 1000), across different families to potentially capture a spectrum of low to high genomic expected values, totaling 3000 samples. At current costs around USD ~6 per sample for DNA extraction and preparation and USD ~24/sample for SNP genotyping, this would amount to USD 90,000. This genotyping cost would be lower, for what regards the selection candidates, in the alternative route of using GS only at the family mean level, or significantly lower if using pooled family genotyping.

The timeline of such a pilot project, for example for a fast growing tropical *Eucalyptus* would be as follows: (1) one year to organize phenotypic data, generate SNP data for the training population, and develop the GS model; (2) two more years to generate and grow open-pollinated (maternal selection only) or five more years for full-sib seedlings (both sexes selection), including, in both routes, SNP genotyping and genomically ranking 1000 selection candidates; (3) four years for cloning, establishing and assessing the predicted versus realized volume growth for a sample of genomically ranked trees in a clonal trial, covering the spectrum of predicted genomic values. This plan would therefore require a minimum of seven to ten years, depending on the route taken. Selected trees could in parallel be top grafted, flower induced to be intermated to generate the parents of the next recurrent selection breeding cycle.

## 8. What Are Some New Outlooks in Genomic Selection in Forest Trees?

### 8.1. Genotyping Cost Reduction by Low-Pass Sequencing and Imputation

Based on the fact that GS is essentially "the animal model with DNA markers", and assuming that tree breeding will continue to be relevant for the foreseeable future, in my opinion, the cost component of SNP genotyping is still the key variable to trigger the full adoption of GS at a larger scale in tree breeding and reach the plateau of productivity in the Gartner hype cycle.

It is still not clear how low should SNP genotyping cost be for a single DNA sample to match or improve over the cost of planting, managing and especially measuring several traits in a progeny or a clonally trialed tree, discounted the value of the wood harvested from the trial. Clearly, GS is expected to significantly cut or even eliminate some phenotyping costs. This is particularly true for expensive measurements of late expressing wood chemical and physical traits, or disease and pest resistance, that require laborious assays. In the case of GS for pest or disease resistance/tolerance, it is important to point out that the challenge of dealing with genetic variation and fluctuations in the pathogen and pest populations is evidently common to conventional breeding and GS. GS does not provide any distinct relief in that respect and it is going to be as challenging or even more challenging than conventional breeding. Comparing the cost of genotyping with the cost of phenotyping, also requires a solid exercise to measure the value of the time savings in accelerating the breeding cycle. Such exercises vary across organizations based on a number of factors, including their cost structure, available personnel teams, breeding strategy, pressure from competition, new products developments and environmental and biotic factors, which in turn impact the aggressiveness of the breeding program. It would be vain to try to discuss this here. Everyone will agree, however, that genotyping cost should always be the lowest possible.

Although SNP arrays are currently the best option, for many breeding programs the cost may still be too high for routine large-scale adoption. The catch is that array manufacture and labor are expensive, with the lower limit in final costs already close to the current selling price, according to current SNP array manufacturers. Genotyping

by sequencing methods using enzyme-based genome complexity reduction were seen as an alternative, but in practice they have not been a robust long-term routine method for highly heterozygous tree genomes. Not only they are not cheaper than arrays anymore, but also, and more importantly, the somewhat irregular sampling and sequencing coverage of loci, results in unacceptable genotype reproducibility and missing data, such that the final effective number of robust markers, after filtering for quality, is much lower than what is expected or advertised [91,92]. Methods based on targeted amplicon sequencing can be cheaper than arrays, provided that very large sample number contracts are made to dilute the initial costs of assay development, and several hundred samples are processed simultaneously to fill up the sequencing flow cell. In other words, it is logistically more difficult to genotype less than a few hundred samples at once. These methods typically interrogate up to 5000 SNPs efficiently, and could prove to be an alternative if such a marker density provides the same prediction ability as array data. For example, a 3000 SNP amplicon sequencing panel was developed for *Cryptomeria japonica* and successfully used for genomic prediction [93].

A potentially interesting alternative to SNP arrays has recently been proposed based on ultra-low-pass whole genome sequencing (0.1–0.5× sequencing coverage) followed by SNP imputation to a comprehensive reference panel. This approach has been around for a while [94], but only became cost efficient in recent years, with the incessant advances in next generation sequencing, and highly multiplexed library preparation methods [95,96]. Low-pass whole genome sequencing plus imputation has already been successfully used for prediction of polygenic risk score in humans [97], and genomic merit in beef cattle [98]. Very strong agreement above 98% was seen between imputed and array-derived SNP genotypes.

There are some potential advantages of this SNP genotyping approach: (1) unlike a fixed SNP content array, the set of variants extracted from the data is flexible, without the need to redesign and manufacture a different array; (2) although variant imputation can also be performed from array data, low-pass sequencing enables less biased imputation than genotyping arrays by not having to pre-stipulate the SNP content; (3) a much larger number of SNPs, in the range of several hundred thousand or millions, can be obtained, and these can be useful to drive GWAS experiments, if desired; (4) the full set of imputed variants from the haplotype reference can capture individual variants in genomic regions where interfering SNPs prevent designing probes for genotyping arrays; (5) it is expected that all, or the vast majority, of the SNPs present in the array will be captured in the low-pass imputed sequence data; the imputed SNP data set can thus be breeder-friendly, tailored to match the currently used array SNP set for seamless data consolidation; (6) the more extensive coverage of variants in the genome might allow including functional variants coming from gene discovery projects into GS models, although whether the inclusion of causative SNPs will lead to substantially increased accuracy of selection is still unclear [99].

This approach requires, however, the upfront construction of a haplotype reference panel used to predict the missing genotypes in the low-pass sequence data. The correct choice of the reference panel influences the imputation accuracy in the genotyped sample. In general, genotype imputation accuracy is higher when the reference panel and the genotyped sample derive from the same or similar population. This should be particularly relevant for high diversity and highly heterozygous tree genomes. Therefore, for the application of GS in tree breeding, the haplotype reference panel should be built by sequencing all the parents of the breeding population to a 15–20× coverage. In the case of *Eucalyptus* and Poplar, with genome sizes in the range of 500–650 Mb, this would be manageable. For example, assume a breeding population involving 100 *Eucalyptus* parent trees with a genome size of 650 Mb. At 20× genome sequencing coverage, this would amount to 1300 Gb of raw sequence data. This project would require preparation of 100 shotgun libraries and sequencing of two S4 flow cell of an Illumina Novaseq 6000 instrument. The total cost of data generation would currently be less than USD 20,000, which is the equivalent to array genotyping less than 1000 samples. With a reference panel in hand, current service providers promise to offer genotyping by low-pass sequencing plus imputation at

around USD 30–40/sample in contracts involving several tens of thousands of samples, a price, however, which is not yet competitive with SNP arrays. Finally, a minor downside of genotyping by low-pass sequencing, is that it requires processing several hundred samples simultaneously to fill up a sequencing flow cell. This might hinder data turnaround when small sets of samples need to be genotyped.

### 8.2. Modern Tree Breeding by Integrating Genomics and Enviromics

A number of new approaches have been developed, largely in crop plants, to exploit the genotype by environment interaction in genomic prediction, showing that the incorporation of multi-environmental models or crop growth models provide better accuracy to predict environment-specific performances when compared to single environment models [58,100,101]. With multi-environmental modelling, GS allows evaluating marker effects and marker effects by environment interactions, ultimately improving prediction accuracy. Moreover, with the integration of crop growth models in the genomic prediction framework, the response of the genotype to the environmental variations allows predicting the performance of selection candidates in untested environments.

Genomic models that take into account multiple traits, multiple environments and their interactions, represent a formidable tool to exploit the correlations between these many variables and better resolve their individual contributions. Moreover, the increasing worldwide availability of geoprocessing technologies and data, now enables the application of the "enviromics" approach. Likewise genotypes at DNA markers, any particular site is characterized by a set of "envirotypes" at multiple "enviromic" markers corresponding to environmental variables that interact with the genetic background, providing informative breeding re-rankings for optimized decisions over different environments [102]. Modern breeding approaches that integrate genomics, phenomics and enviromics data and models [103] have not yet been reported in forest tree breeding but undoubtedly will be in the near future. They represent a huge potential to predict the performance of genetic material across all sites of interest, including untested ones in present environmental conditions as well as currently untestable sites predicted by climate change models.

### 8.3. Phenomic Selection

In the last few years, a number of studies have proposed phenotype prediction in breeding by shifting from using DNA markers data to some alternative form of endophenotype data such as transcriptomics, metabolomics [104,105] or near infrared reflectance spectroscopy (NIRS), this latter case, coined "phenomic selection" (PS) [106]. While the relatively low throughput and high costs of transcriptomics and metabolomics data collection might not be a reasonable alternative to SNP data at current costs, NIRS is significantly cheaper with potentially equivalent throughput by modern online systems [107]. PS would be carried out essentially as GS, but instead of collecting SNP genotypes, NIRS spectra are collected. Most studies to date comparing PS with of GS, were carried out in cereal grains, but one included Poplar [106]. Recently Robert at al. [108] reviewed a comprehensive set of selected publications that used NIRS to predict agronomic traits and proposed different promising applications of PS in breeding. In these studies, NIRS data were measured on specific plant tissues and used to build a relationship matrix in substitution to the conventional pedigree matrix and the predictive abilities obtained were generally similar to those obtained with GS.

The fundamental premise of PS by NIRS, as discussed by Rincent et al. [106] in their seminal paper, is that the NIRS spectra of a sample is largely related to its chemical composition, which itself is a result of endophenotypes and genetics. NIRS data is therefore expected to explain at least a portion of the genetic variance, and as a result, be able to predict traits unrelated to the analyzed tissue or in independent environments. The other premise for PS to work as GS is that NIRS data can capture relationships at an equivalent granular level as genomic data, therefore explaining the Mendelian sampling term within families. Studies that used NIRS on eucalypt wood samples to estimate

genetic parameters for chemical and physical properties [49], or on leaf samples to study population differentiation for plant metabolites [109], showed that most NIRS wavelengths display genetic variability and may differentiate full-sibs, although no specific analyses were done in that respect. In the only study comparing PS with GS in a forest tree, NIRS data were collected on pulverized wood from stem sections collected on two-year-old Poplar trees and compared to genomic data for 7,808 SNPs. The efficiency of GS and PS was estimated to predict the performance of new individuals within a cross-validation framework using GBLUP for a number of traits in different environments. PS showed an equal or better performance than GS for height and circumference growth in both sites but a worse performance for bud flush, bud set and rust resistance [106].

NIRS is a familiar tool to tree breeders and has been commonly used for a few decades, especially to indirectly evaluate wood chemical properties following the construction of a prediction model [110]. Equipment can be acquired and operated at reasonable costs. Costs of sample preparation in the lab compared to genotyping would probably still be cheaper, provided that labor costs are not too high. Alternative hand-held equipment could potentially be used to collect spectra in the field or nursery, making the process extremely easy and even cheaper. However, there are at least two main questions to be addressed by future research, before this approach may be considered as a potential alternative to GS. As pointed out earlier [106,108], to match GS for ultra-early indirect selection, NIRS data have to be collected on young seedling tissues and these tissues and their spectra have to contain the relevant endophenotype information to predict late expressing growth and wood properties, in other words, predict the phenotypes of different tissues and different ages. One possibility is to collect NIRS spectra on the training population at the same age, the same tissue and same controlled environment as the NIRS spectra will be collected on the selection candidates, evidently provided that the relevant final phenotypes (e.g., growth and wood traits at harvest age) are also measured on the training population. Second, being it a phenotype, a NIR spectrum contains both additive, non-additive and genotype x environment effects. It is not yet clear what proportion of the variance in NIR spectra is effectively heritable and therefore passed across generations in a breeding program. Well controlled growing conditions in the nursery could facilitate the obtention of NIRS spectra with less environmental impact, i.e., higher heritability, reflecting more of the genetics and thus likely providing better predictions. In summary, there are still many open questions about the true applicability of PS as an alternative to GS. However, if the NIRS data is able to correctly capture the genetic relationships as efficiently as SNPs data, and if the relationships explain the majority of the predictive ability, NIRS-based PS might work as well as SNP-based GS, and likely at a significantly reduced cost in a logistically simpler way in the organization. Research in PS using NIRS data is warranted in forest tree breeding, especially thinking about species and breeding programs that will not afford SNP genotyping costs anytime soon.

## 9. Conclusions: GS Is "Lindy" and Mirrors the "Buy the Whole Haystack" Concept

In concluding this perspective piece, if all the experimental data already available not only in forest trees but also in crops and domestic animals, are not sufficient to convince the reader that we are climbing the slope of enlightenment of marker assisted forest tree breeding, there are two more ideas that might help. In my reflections about GS, two thought-provoking concepts coming from other fields, support the expected long-term value, robustness and efficiency of GS as a tree breeding tool for years to come.

The first one is the "Lindy effect", a theorized phenomenon by which the future life expectancy of a technology, a product, a book or an idea is proportional to its current age [111,112]. Because the underlying foundation of GS are the "Lindy" century-old proven quantitative genetics concepts of the infinitesimal model and resemblance among relatives, it should also resist obsolescence and have a longer remaining life expectancy. Additionally, by the same proxy to quantitative genetics, GS could be seen as having a long history of validated data and impact in plant breeding, different from the vast majority of the

ephemeral biotechnology neomania bandwagons proposed, that have not survived the hype cycle in the last thirty years. Therefore, the breeder should not have any fud (fear uncertainty and doubt) as far as the fundamental concepts underlying GS. Nevertheless, it is important to remind that the role of quantitative genetics in modern plant breeding has evolved. As pointed out by Rex Bernardo [113], the focus now is much more on identifying which candidates have the best genotypic value for one or more continuous traits, in a set of environments, and less on partitioning variances or estimating heritabilities. Breeding by genomic prediction using SNP marker data and mixed-model approaches, has therefore become much more computational and empirical, precluding many assumptions of classical quantitative genetics.

The second concept to which GS conforms particularly well, comes from the world of investing. Instead of trying to guess which individual stocks or active fund manager, among the myriad available, will provide the best returns-a practice that in over 85% of the times results in underperforming the market in the medium to long term (see SPIVA accessed on 29 July 2022 at https://www.spglobal.com/spdji/en/research-insights/spiva/)-it is much more efficient and humble to "buy the whole market" through a broad-based index fund that captures the full effects of the market [114]. Or, in other words, as John Bogle's famous quote, instead of trying to find a needle in the haystack–or a QTL or a gene in the genome, for that matter, just buy the whole haystack, the needle will be there [115]. With GS, the breeder in effect uses the same fundamental principle. Because we do not know what are the genes that really make a tree grow or make better wood, instead of trying to search for individual genes or QTLs, GS efficiently captures the whole spectrum of genetic effects and their interactions contributed by the entire genome. By doing that, all the relevant "effects" and their complex interactions are likely to be in the predictive model.

This is not to say that there is no value in trying to understand the individual mechanistic components underlying the complexity of quantitative traits. It is a very noble and challenging scientific enterprise, although I doubt that such contributions to operational tree breeding will pay off anytime soon. I just contend that, differently from crops that have a much older history of domestication, directional selection and genetical research, and enjoy key experimental resources such as inbred lines, haploid induction system and well characterized mutant collections, to name a few, we still lack those experimental tools to carry out such a bold endeavor effectively in genetically heterogeneous, largely undomesticated forest trees. Among such experimental resources, the development of inbred lines is probably the most important one and, in my opinion, it should be prioritized to advance mechanistic research in forest trees. As Mendel realized early in his research, inbred lines allow much more precise experimental predictions, interpretations and validations, now expanded from genetics to genomics and molecular biology. Furthermore, inbred lines will open major opportunities for parental line mate selection and genomic prediction of untested hybrids for full exploitation of non-additive variance. In tropical *Eucalyptus* this approach has been a major focus of our collaborative research with forest based industrial partners in the last ten years. By advancing lines by self-pollination of all elite parents of a breeding program, accelerated by genome-wide selection for maximal conversion to SNP homozygosity, coupled with early flowering by top grafting of inbred seedlings, $S_3$ lines with close to 90% converted homozygosity have been generated (Dr. Elizabete Takahashi, personal communication).

As we climb the slope of enlightenment, GS will soon reach the plateau of productivity in several forest tree breeding programs. As more organizations adopt this alternative breeding approach, SNP genotyping will become even more accessible or possibly genomic data will be substituted or complemented by some form of endophenotype data such as NIRS. Yet predictive abilities are impacted by GxE interaction and driven mainly by relationships, such that population-specific predictive models will always be necessary. While the fundamental aspects of GS in tree breeding are now firmly established, strategic, logistics and financial aspects for the optimized adoption of GS on a case-by-case basis

should now be the emphasis of research to fully integrate this genomic breeding method into routine tree improvement.

**Funding:** This work was supported by FAP-DF grants NEXTREE 193.000.570/2009 and NEXTFRUT 0193.001198/2016, and a CNPq fellowship productivity grant 306866/2018-8.

**Institutional Review Board Statement:** Not applicable.

**Informed Consent Statement:** Not applicable.

**Data Availability Statement:** Not applicable.

**Acknowledgments:** I thank Orzenil B. Silva-Junior for calling my attention to the new developments of low-pass sequencing for SNP genotyping.

**Conflicts of Interest:** The author declares no conflict of interest.

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
