# Peer review of "Twelve Years into Genomic Selection in Forest Trees: Climbing the Slope of Enlightenment of Marker Assisted Tree Breeding"

_forests, doi:10.3390/f13101554_

Round 1

Reviewer 1 Report

Overall, I found the paper to be quite useful and informative, both for tree breeding specialists and for researchers whose expertise lies just outside the field of genomic selection (like myself). Indeed, this paper greatly helped me to clarify differences between MAS, GS and various QTL applications.  Coverage of GS in forest trees, comparisons and advantages of GS compared to conventional breeding, and factors that impact its success is extremely thorough and authoritative. Practical considerations are also carefully considered, and Figure 2, which diagrams a step-by-step plan for a GS pilot project (complete with timeline and cost estimates), provides pragmatic guidance rarely found in academic literature.  Finally future directions which includes low-pass sequencing, enviromics, and phenomic selection provide a thought-provoking window to the future.

I was not familiar with “Gartner’s hype cycle” nor the term “slope of enlightenment. I’d suggest explaining this term and concept in the abstract, at least briefly, rather than later in the introduction. That said, I really like Figure 1, and although some of the timing and labels are debatable (as the author acknowledges), the overall metaphor is clever and useful.

The writing style of this manuscript is somewhat informal and the author uses creative analogies to illustrate some complex topics. While I would choose different wording in places, the paper is well written overall and references are comprehensive. I noted several minor typos and missing words scattered about, so a thorough proof-reading would be beneficial. There are a lot of abbreviations and acronyms used (by necessity), so a table of these for readers to refer to would be useful.

Reviewer 2 Report

Overall, this is a useful contribution to the special issue, and makes a clear case for the value of genomic prediction in tree breeding programs that meet the conditions outlined in the text. The manuscript states clearly that it represents the author's personal perspective based on his experience with tropical species, and some aspects of the text reflect that.

In my view, lines 310 to 312 misstate the relationship between predictive accuracy and other factors. The statement is "Predictions across populations will be low or zero due to differences in allele frequencies, LD pattern, haplotypes and population-specific allele effects, but mainly because family relationships are absent." It is well-known that in Holstein cattle (for example), GS works just fine in the absence of known pedigree relationships because there are few differences in the listed factors within the global Holstein population. It is the genetic factors that are important, not the presence or absence of known family relationships.

The manuscript seems to assume that "shortening the breeding cycle" faces few if any biological obstacles. Flower induction is mentioned frequently, either by top-grafting or other unspecified methods - this works very well for some species, but others are recalcitrant and reproductive maturity cannot be accelerated so easily. The time from flowering to seed, and from seed harvest to plantable propagules, are other biological variables that will have a significant impact on the potential to reduce the length of the breeding cycle. Another assumption is that pest and disease resistance traits, cited as among those difficult to measure and therefore most likely to benefit from GS, depend only on the genetic variation of the host tree. Unfortunately this is not the case - pest and disease populations have their own genetic variation and response to selection, and will adapt to genetic changes in the host population. Continued phenotypic measurements of disease and pest resistance are likely to increase in importance, particularly in temperate and boreal regions affected by climate change and concomitant shifts in pest and disease pressures faced by forest stands.

Reviewer 3 Report

Review of the paper entitled “Twelve years into genomic selection in forest trees: climbing the slope of enlightenment of marker assisted breeding”, written by Dario Grattapaglia and submitted to Forest (1866831).

Thanks to his long experience in both conventional and molecular tree breeding, the author describes in details genomic selection and compares it to other molecular breeding approaches (marker-assisted selection) that have been developed over the past decades. He presents the development of these molecular methods in the context of tree improvement programs using the concept of hype cycle curve, introduced by Gartner Inc. in 1995. He argues that genomic selection, along this curve, is in the slope of enlightenment section and is ready to be adopted by tree breeders. To support his argument, he lists the main advantages of genomic selection as a breeding method in forest trees when compared to conventional breeding. He also describes the factors that are critical to make genomic selection a successful breeding method. To help tree breeders who have not yet implemented genomic selection in their breeding program, he presents a step by step plan of a simple pilot project to evaluate and demonstrate the potential of genomic selection in a recurrent selection breeding program. In a last section, he provides the reader with his perspectives on future developments in genomic selection and other related methods.

I enjoyed reading this paper. I am convinced that it will be of interest not only to tree breeders who have not yet introduced genomic selection in their breeding programs and who are thinking to do so, but also to those who have already some experience with genomic selection. The manuscript is well written and I have only few minor comments/suggestions.

1-      Table 1, When it works: I would also say that it works better when the effective population size is small (under 100).

2-      L. 212: With regard to the breeding equation, I would suggest that the g symbol should represent the ratio of genotyping cost vs. conventional breeding cost. Alternatively, to have a valuable comparison with conventional breeding, a g value should be estimated for both methods, i.e., genomic selection and conventional breeding and compared.

3-      L. 258-260: If the author wants to support this statement with a reference, I would suggest Beaulieu et al. (2022) Sci. Rep. 12: 3933.

4-      L. 262: I would suggest adding one or two references here. One of them for a polymix trial strategy could be Lenz et al. (2020) Heredity 124: 562-578.

5-      L. 518: Check the sentence.

6-      L. 520: were included instead of ere included.

7-      L. 545-553: Although it was not specifically indicated in Chamberland et al. (2020), I would like to add that using records of existing costs of conventional tree breeding, we have estimated that the cost of genomic selection was similar to that of conventional breeding, that is about $5 CDN/hectare. Of course, that value is dependent on the total number of hectares of plantation on which the costs are amortized over years. Nevertheless, this cost is very small and almost not significant as compared to other costs related to plantation (plant production, site preparation, planting, tending) which can sum up to $2000 to $3000 per hectare.

8-      L. 565-571: Ideally, the breeder should have access to an existing genotyping chip to obtain a reasonable genotyping price per sample. Otherwise, a small-scale project could be much expensive.

9-      L. 680-686: I fully agree with this comment.  Most of genomic selection study results obtained using GbS do not agree with the general trends observed with studies using genotyping arrays or exome capture.
